# The Impacts of Cholesterol, Oxysterols, and Cholesterol Lowering Dietary Compounds on the Immune System

**DOI:** 10.3390/ijms232012236

**Published:** 2022-10-13

**Authors:** Rintaro Yanagisawa, Chaoqi He, Akira Asai, Michael Hellwig, Thomas Henle, Masako Toda

**Affiliations:** 1Laboratory of Food and Biomolecular Science, Graduate School of Agricultural Science, Tohoku University, Sendai 980-8572, Japan; 2Department of Endocrinology, Diabetes and Metabolism, Graduate School of Medicine, Nippon Medical School, Tokyo 113-8603, Japan; 3Chair of Special Food Chemistry, Technische Universität Dresden, 01062 Dresden, Germany; 4Chair of Food Chemistry, Technische Universität Dresden, 01062 Dresden, Germany; 5Institute of Fermentation Sciences, Faculty of Food and Agricultural Sciences, Fukushima University, Fukushima 960-1296, Japan

**Keywords:** cholesterol, oxysterol, immune modulation, inflammation, β-glucan, plant stanol/sterol, omega-3 lipids, polyphenol, soy proteins

## Abstract

Cholesterol and its oxidized forms, oxysterols, are ingested from foods and are synthesized de novo. Cholesterol and oxysterols influence molecular and cellular events and subsequent biological responses of immune cells. The amount of dietary cholesterol influence on the levels of LDL cholesterol and blood oxysterols plays a significant role in the induction of pro-inflammatory state in immune cells, leading to inflammatory disorders, including cardiovascular disease. Cholesterol and oxysterols synthesized de novo in immune cells and stroma cells are involved in immune homeostasis, which may also be influenced by an excess intake of dietary cholesterol. Dietary compounds such as β-glucan, plant sterols/stanols, omega-3 lipids, polyphenols, and soy proteins, could lower blood cholesterol levels by interfering with cholesterol absorption and metabolism. Such dietary compounds also have potential to exert immune modulation through diverse mechanisms. This review addresses current knowledge about the impact of dietary-derived and de novo synthesized cholesterol and oxysterols on the immune system. Possible immunomodulatory mechanisms elicited by cholesterol-lowering dietary compounds are also discussed.

## 1. Introduction

Cholesterol, a vital lipid, is widely distributed throughout the body. It plays a crucial role as a major constituent of cell membranes, precursor of sex hormones, corticosteroids, bile acids, and vitamins, and is involved in cellular metabolism and signaling pathways [1,2]. Oxysterols, the oxidation products of cholesterol, are also involved in numerous cellular events [3,4,5]. Cholesterol and oxysterols are ingested through food (exogenous) and synthesized de novo (endogenous) and are vital for the immune system due to their involvement in differentiation, migration, and the functioning of immune cells [6]. Both dietary and the de novo synthesis of cholesterol and oxysterols are also linked to multiple inflammatory disorders, including atherosclerosis [1,4]. Several oxysterols synthesized de novo can potentially be used as disease- and tissue-specific inflammatory biomarkers [7,8].

The ability of dietary compounds such as β-glucan, plant sterols, omega-3 fatty acids, polyphenols, and soy proteins, to lower cholesterol has been investigated in attempts to reduce cholesterol absorption and biogenesis. The United States Food and Drug Administration (USFDA), the European Food Safety Authority (EFSA), and food safety authorities elsewhere have approved oat/barley β-glucan and plant sterols/stanols as dietary components with lowering effect on blood cholesterol [9,10,11,12,13,14]. Other dietary compounds with the potential of cholesterol lowering effects include omega-3 fatty acids, polyphenols, and soy proteins. Notably, such dietary compounds might even modulate immunity [15,16,17,18,19,20,21]. The diverse action mechanisms of these immunomodulatory molecules have been identified. This review introduces how dietary and de novo synthesized cholesterol and oxysterols affect cellular and molecular events in immune cells; the main participants include macrophages, dendritic cells (DCs), innate lymphoid cells (ILCs) in innate immunity, and T cells and B cells in adaptive immunity. In addition, the potentials of cholesterol lowering dietary compounds in immune modulation are also discussed.

## 2. Intake and De Novo Synthesis of Cholesterol and Oxysterols

### 2.1. Dietary and De Novo Synthesized Cholesterol

#### 2.1.1. Dietary Cholesterol

Both dietary and de novo synthesized cholesterol contribute to the maintenance of overall cholesterol homeostasis. The major sources of dairy cholesterol are animal products, such as meat (including poultry and seafood), egg yolk, and dairy products [22]. As physiological cholesterol levels are mutually regulated by the endogenous and exogenous pathways of cholesterol metabolism, the amount of dietary cholesterol has only a modest impact on circulating cholesterol levels in the general population. However, the amount of cholesterol intake can be a non-negligible factor in the rise in low-density lipoprotein (LDL) cholesterol and oxysterol levels, particularly in populations with a high risk of cardiovascular disease (CVD), e.g., in patients with dyslipidemia and diabetes [23].

#### 2.1.2. De Novo Synthesis of Cholesterol

The liver is the main site for cholesterol metabolism and storage. In the liver and other organs, e.g., intestinal tissues, cholesterol is synthesized de novo through the mevalonate-isoprenoid pathway (Figure 1). A series of isoprenoid products of this pathway play vital roles in multiple cellular functions, including protein post-translational modifications, cell signaling, cell membrane integrity, cell cycle progression, and cholesterol synthesis. Cellular cholesterol level is tightly regulated by repression system via inhibition of the sterol regulatory element-binding protein (SREBP) 2 through the binding of cholesterol with SREBP-cleavage-activating protein (SCAP)/insulin-induced gene 1 (INSIG1), and degradation of hydroxymethylglutaryl-coenzyme A (HMG-CoA) reductase through ubiquitylation [1].

SREBP-2 forms a complex with SCAP in the endoplasmic reticulum (ER) membrane (Figure 2). When the levels of intracellular cholesterol are low, SREBP-2/SCAP complex is transported to the Golgi apparatus, where it undergoes a two-step processing by the Golgi-localized proteases Site-1 protease(S1P) and Site-2 protease(S2P) to release the NH2-terminal fragment of SREBP-2 (nSREBP-2) [1,2]. nSREBP-2 is translocated into the nucleus and binds to the sterol regulatory elements (SREs) in the promoters, that induces the expression of genes related to the mevalonate pathway, such as HMG-CoA and other key enzymes, increasing the levels of cholesterol. When the levels of intracellular cholesterol are high, the binding of cholesterol to SCAP alters its structure and SCAP forms a complex with INSIG1. This causes the SREBP-2/SCAP complex to be an inactive form in the ER membrane. Oxysterols bind INSIG and form the complex of SREBP-2/SCAP/INSIG1 that prevent SREBP-2 cleavage and promote the ubiquitination and degradation of HMG-CoA [1,2].

All the cholesterol needed by the human body can be synthesized throughout childhood and the adult years; however, in most instances, exogenous cholesterol from foods is also consumed in addition to endogenously synthesized cholesterol. The intestinal cholesterol absorption rate is estimated to vary from approximately 25% to 85% in the amount of dietary cholesterol intake, but is approximately 50% on average [24].

### 2.2. Dietary and De Novo Synthesized Oxysterols

#### 2.2.1. Dietary Oxysterols

Cholesterol gives rise to various oxysterols via enzymatic and non-enzymatic routes [25]. Dietary oxysterol is produced by the oxidation of cholesterol with oxygen and the reactive oxygen species (ROS) during cooking and storage of foods. Several studies showed that oxysterols are increased under conditions of microwave cooking or long-term frozen storage [26,27,28,29]. The main sources of dietary oxysterols are processed or stored in eggs and dairy products, and other cholesterol rich foods. Cholesterol oxidation products can already be detected in raw meat (2–11 mg/kg fat), and the concentrations rise during thermal processing (3.4–35 mg/kg fat). Egg yolk powders (441–714 mg/kg fat) and butter, as well as butter oil (13.7–27.3 mg/kg fat), are major dietary sources of cholesterol oxidation products [26,27,28,29]. The most commonly detected individual oxysterols in foods are C7 cholesterol oxides: 7α-hydroxycholesterol (7α-HC), 7β-hydroxycholesterol (7β-HC), and the oxidation product of both, 7-ketocholesterol (7-KC) (Figure 3). 7β-HC is generated only by non-enzymatic oxidation, whereas 7α-HC is generated by enzymatic and non-enzymatic oxidation. In addition to the C7 cholesterol oxides, epoxidation products, 5α,6α-epoxycholesterol (5α,6α-EPOX) and 5β,6β-epoxycholesterol (5β,6β-EPOX), are generated by non-enzymatic oxidation [30]. These epoxides are less important in milk, but comparatively abundant in egg fat, where they can be the main species of cholesterol oxidation products [29]. Storage mainly leads to an increase in sterol oxidation. Much more strongly, the contents of 7α-HC and 7β-HC rise during the storage of yolk powder [29].

#### 2.2.2. De Novo Synthesized Oxysterols

Oxysterols are de novo synthesized from cholesterol by enzyme belonging to the cytochrome P450 family [31,32]. The major oxysterol synthesized in the liver and other organs is 27-hydroxycholesterol (27-HC) (Figure 3), which has a positive correlation with plasma cholesterol levels [33]. 24(S)-hydroxycholesterol (24S-HC) is highly synthesized in the brain. In the plasma of healthy adults, the levels of cholesterol are in the order of 1 to 3 mg/mL, whereas the most detectable oxysterol is 27-HC at a concentration of a few ten of ng/mL, followed by 24S-HC, 25-hydroxycholesterol (25-HC), 7-KC, 4β-hydroxycholesterol (4β-HC), 7α-HC, and 7β-HC and others to the lesser contents [34]. 27-HC can pass through the blood brain barrier and influx to the brain, whereas 24S-HC effluxes from the brain to the peripheral circulation.

Diet and pathological condition alter oxysterol levels in tissues, blood, and cells [35,36,37]. Feeding animals with a high-cholesterol diet increases the serum levels of several oxysterols, including 27-HC and 4β-HC, and the brain tissue levels of 24S-HC [38,39]. Several studies have suggested an involvement of 24S-HC and 27-HC in the pathogenesis of Alzheimer diseases [40,41]. Patients with hypercholesterolemia, or type 2 diabetics, increase concentrations of plasma 7-KC, that could be due to hyperglycemia enhancing cholesterol oxidation by free-radical-mediated pathways [37,42]. In atheroma plaque samples from patients undergoing carotid endarterectomy, approximately 437 ng/mL of 27-HC and 44 ng/mL of 7-KC, and to a lesser extent 7α-HC, 7β-HC, 24S-HC and 25-HC, were detected as the mean concentrations of the samples [43]. In oxidized low-density lipoprotein (ox-LDL), 7-KC was the most abundantly detected oxysterol, followed by 25-HC, 7α-HC, and 5α,6α-EPOX [44]. In human peripheral blood mononuclear cells (PBMC), abundant 7-KC and lesser levels of 25-HC were detected [36].

Importantly, 25-HC is de novo synthesized at low levels in the ER of all cells due to the expression of cholesterol 25-hydroxylase in the cellular compartment [45]. This physiological synthesis of 25-HC is involved in the synthesis, intracellular transport, and the storage of cholesterol [45,46]. 25-HC promotes activity of acyl-CoA cholesterol acyltransferase (ACAT) to the esterification of cholesterol and represses nSREBP2-induced genes, that regulate cholesterol availability in the cells [46]. 27-HC, an endogenous oxysterol produced from cholesterol by cholesterol 27-hydroxylase (CYP27A1), also regulates intracellular cholesterol homeostasis [32,33]. However, among oxysterols, the synthesis of 25-HC is highly promoted in macrophages and dendritic cells (DCs) in pathogen infection and is engaged in multiple processes in intrinsic immunity (see Section 3.6) [45,46]. 25-HC is also involved in T-cell and B-cell differentiation (see Section 3.3 and Section 3.4) [45,46].

Oxysterols are also generated in mitochondria. Cholesterol levels in microcoria membrane is lower than plasma and ER membranes [47]. However, mitochondria are a main source of cellular ROS due to respiration, and induces the auto-oxidation of cholesterol [47,48,49]. In addition, CYP27A1 is expressed in the inner membrane of mitochondria [49]. Several oxysterols have been detected in mitochondria, that include an auto-oxidation product of cholesterol, 7-KC, and de novo synthesized oxysterols, 24-HC, 25-HC, and 27-HC [35]. Bile acids are synthesized from cholesterol via two pathways: the classic (primary) pathway; and the alternative (acid) pathway. Mitochondrial CYP27A1 is involved in the production of bile acid by hydroxylating cholesterol to form 24-HC, 25-HC, and (25R)-26-hydroxycholesterol (26-HC) in the acid pathway, whereas microsomal 7α-hydroxylase (CYP7A1) generates 7α-HC from cholesterol in the first rate-determining step of the classic pathway [49]. The role of oxysterols in mitochondria function is still not well known.

## 3. The Impact of Cholesterol and Oxysterols on Immune Cells

Cholesterol and oxysterols influence the molecular and cellular events of immune cells and subsequent biological responses through multiple mechanisms: (i) activating pattern recognition receptor-mediated cascades [50]; (ii) triggering NLRP3 inflammasome-mediated cascades [51]; (iii) inducing apoptosis by ROS production [52]; (iv) interacting with nuclear receptors such as the liver X receptor (LXR)α/β, retinoic acid receptor-related orphan receptor (ROR) α, RORγt, or estrogen receptor α (Erα) [53]; (v) mediating chemotaxis through oxysterol-binding G-protein coupled-receptor 183 (GPR183, also known as Epstein-Barr virus-induced molecule) [54]; and (vi) influencing cholesterol/oxysterol-related metabolic and signaling pathways [55,56]. This section describes the effects of cholesterol and oxysterols on immune cells and their associated mechanisms.

### 3.1. Macrophages

#### 3.1.1. Activation and Induction of Inflammation

Macrophages are differentiated from monocytes and play a crucial role in innate immunity. The impact of cholesterol and oxysterols on monocytes and macrophages in the development of atherosclerosis has been intensively investigated. Elevated LDL cholesterol levels are directly associated with the risk of atherosclerotic cardiovascular events [57]. LDL can be modified to ox-LDL, which causes cholesterol accumulation in the coronary artery, thereby triggering deposition, inflammatory response, and the apoptosis of monocytes and macrophages [58].

Ox-LDL binds to several pattern-recognition receptors (PRRs), including Toll-like receptor (TLR) 4, and scavenger receptors, including scavenger receptor class A (SR-A), class B (SR-B), CD36, and lectin-like oxidized LDL receptor-1 (LOX-1) [59,60]. The engagement of these PRRs trigger the production of pro-inflammatory cytokines, interleukin (IL)-6, and tumor necrosis factor-α (TNF-α), through the activation of the nuclear factor κB (NF-κB) and mitogen-activated protein kinase (MAPK) signaling pathways in monocytes and macrophages [61,62]. Endocytosed ox-LDL by PRRs form crystalized cholesterols in the lysosome compartments, which trigger the production of pro-inflammatory IL-1β through formation of the inflammasome [63].

Oxysterols are contained in ox-LDL and ox-VLDL and those released in cells and accumulated in tissues also activate monocytes and macrophages and induce inflammation. 7-KC, 25-HC, and 27-HC, reportedly act as ligands for the nuclear receptors RORα, RORγ, and ERα, that leads to the expression of pro-inflammatory cytokines and chemokines in cultured macrophages and monocytes [5,58,64]. Among the products of enzymatic and non-enzymatic oxidation of cholesterol, 27-HC and 7-KC highly circulate in the blood stream, respectively [37,42,52]. Immunological properties of 25-HC, 27-HC, and 7-KC, have been extensively investigated in monocytes, macrophages, and many other cells [52,65,66,67,68,69].

#### 3.1.2. Induction of Anti-Inflammation

Several oxysterols including 27-HC, 25-HC, 24S-HC, and 24(S),25-epoxy cholesterol (24,25-EPOX) are ligands of LXRs [5,69]. LXRs regulate cholesterol metabolism by inducing expression of ATP-binding cassette protein A1 (ABCA1) and ABCG1, mediating the efflux of cholesterol, together with transcriptional factors, SREBP1c, and carbohydrate-response element-binding protein (ChREBP). LXRs also are engaged in anti-inflammatory pathways to suppress the expression of TLR-mediated pro-inflammatory cytokines and signals in macrophages. Dang et al. showed that 25-HC inhibits the activation of SREBP by binding LXRα and cholesterol synthesis, that suppresses inflammasome formation [70]. However, several studies also shown that 25-HC exhibits pro-inflammatory properties [71,72]. A discrepancy in the effects of 25-HC might be explained by the fact that oxysterols bind several nuclear receptors, in addition to LXRs.

Recent studies showed that 25-HC and 25-hydroxycholesterol 3-sulfate (25HC3S) are epigenetic regulators by acting as ligands of DNA methyltranferase-1 (DNMT-1) [73]. 25HC3S is formed from 25-HC by sulfotransferase 2B1b (SULT2B1b). Wang et al. found that 25-HC activates DNMT-1 up to 8-fold, whereas 25HC3S inactivates DNMT-1, 3a, and 3b [74]. The same research group showed that high glucose increases CpG methylation in promoter regions via increasing nuclear 25-HC levels, which silences key gene expressions involved in MAPK-ERK, calcium-AMPK, and insulin secretion signaling pathways, and type II diabetes mellitus in hepatocytes, whereas 25HC3S increases gene expressions by demethylating 5mCpG in these promoter regions [74]. As synthesis of 25-HC is promoted in macrophages and DCs upon pathogen infection, similar epigenetic regulation by 25-HC and 25HC3S could occur in the cells.

24S-HC, an oxysterol highly detected in brain and microglia (a macrophage population in the central nervous system), is capable of activating silent information regulator-1 (SIRT1), a nicotinamide adenine dinucleotide (NAD)-dependent deacetylase by inducing a cell redox imbalance [75,76]. SIRT1 can inhibit gene expression of pro-inflammatory molecules through histone deacetylation, suggesting that 24S-HC could have anti-inflammatory property [77]. Taken together, several oxysterols modulate inflammation by epigenetic and epigenomic modification.

#### 3.1.3. Differentiation

Depending on the local microenvironment, macrophages are differentiated into classically activated (pro-inflammatory M1) or alternatively activated (anti-inflammatory M2) phenotypes. Few studies have provided direct evidence that cholesterol and oxysterol skew M1 or M2 macrophage differentiation. Buttari et al. showed 7-KC induces pro-inflammatory responses in both M1 and M2 macrophages though NF-kB activation [78]. It is necessary to assess whether cholesterol and oxysterol skew macrophage M1 differentiation, or if they only transiently induce pro-inflammatory properties.

#### 3.1.4. Migration

Upon activation e.g., via TLR4 engagement, macrophages produce 25-HC and 7α,25-dihydroxycholesterol (7α,25-HC), a metabolite of 25-HC catalyzed by two enzymes, CH25H and CYP7B1 [79,80]. Stromal cells within lymphoid organs also produce 7α,25-HC [56]. 7α,25-HC is a chemoattractant binding to GPR183. Macrophages and stromal cells coordinate migration of B cells, T cells, and DCs in secondary lymphoid tissue by producing 7α,25-HC (see Section 3.2, Section 3.4 and Section 3.5) [53]. Such coordination of immune cell trafficking is crucial for efficient induction of immune response and formation of microstructure in the secondary lymphoid tissues.

### 3.2. Dendritic Cells

#### 3.2.1. Activation

DCs are professional antigen-presenting cells (APCs) for the induction of optimal T-cell immunity. De novo synthesized 25-HC and ox-LDL enhance the T-cell stimulatory capacity of human monocyte-derived DCs by enhancing the expression of co-stimulatory molecules (CD80, CD83, and CD86) and HLA-DR, and pro-inflammatory cytokine production in human monocyte-derived DCs [81].

Importantly, 25-HC also contributes to the defense against virus infection by APCs. Plasmacytoid DCs (pDCs) produce high amount of type I IFN (IFN-α and IFN-β) upon the recognition of virus components with TLR3, TLR7/8, and many other PRRs. Type I IFN binds to the Type I IFN receptor, which is expressed in most cells in humans and rodents, and triggers 25-HC production by inducing gene expression of cholesterol 25-hydroxylase, in addition with induction of many other antiviral molecular responses (see Section 3.6.) [82,83,84,85].

#### 3.2.2. Migration

DC migration in the secondary lymphoid organs is coordinated by the axis of CCR7 and its ligands, CCL19 and CCL21, which are constitutively expressed by peripheral lymphatic endothelial cells and lymph node stroma cells [56]. In addition, GPR138 and 7α,25-HC induce DC migration to secondary lymphoid organs. Mouse splenic DCs highly express GPR138 and migrate in response to 7α,25-HC in the T-cell area of spleens, whereas 7α,25-HC deficiency reduces the frequency of splenic CD4^+^ DC number in mice [81,86,87].

#### 3.2.3. Differentiation

25-HC promoted expression of CD11c, a DC marker in bone marrow-derived murine DCs (BMDCs), suggesting that oxysterol promotes BMDC differentiation [88,89]. LXRs and ERα were identified as receptors for the effect of 25-HC on the DC differentiation [88,89]. 25-HC is detected in plasma [33]. It would be crucial to investigate whether and how blood cholesterol and oxysterol levels influence on DC differentiation and function.

### 3.3. T Cells

#### 3.3.1. Activation

T-cell activation is initiated by the binding of the T-cell receptor (TCR)–CD3 complex to a foreign peptide bound to an MHC molecule presented on APCs. Upon the binding, TCRs form cluster in membrane lipid rafts and initiate signaling for activation. Cholesterol, a key component of lipid membrane, is critical for TCR clustering and signaling, and T-cell function [90]. The importance of cholesterol in TCR clustering and signaling has been verified in experimental systems using CD8^+^ T-cells. Increased membrane cholesterol in CD8^+^ T-cells by pharmacologic and genetic blockades of ACAT1 improves TCR clustering and signaling, that enhances proliferation and killing function in murine tumor models [91]. Kidani et al. showed that SREBP2-mediated cholesterol synthesis is dispensable for proliferation and effector function of CD8^+^ T-cells [92]. LXR-mediated cholesterol efflux suppressed T cell activation and proliferation, suggesting the importance of cholesterol in the cellular events [93].

#### 3.3.2. Exhaustion

T cells fall into a state of dysfunction, referred to as T-cell exhaustion, during many chronic infections and cancer. Tumor tissues tend to contain high cholesterol content [94]. Ma et al. showed that CD8^+^ T-cells obtain cholesterol from tumor tissues and thereby promote T-cell exhaustion, that accompanied by the expression of immune co-inhibitory receptors, PD-1, 2B4, TIM-3, and LAG-3 [95]. Cholesterol and oxysterol metabolism in tumor infiltrating T-cells appears to be influenced by exogenous cholesterol from the tumor environment, affecting cellular function negatively [96]. It is crucial to investigate whether high cholesterol diet is a risk factor influencing cancer, leading to such T-cell dysfunction and immune suppressive condition.

#### 3.3.3. Migration

T follicular helper (Tfh) cells are located in the secondary lymph nodes, and play a crucial role to elicit B-cell proliferation and differentiation into plasma cells. The axis of 7α,25-HC with GPR183 induces migration of Tfh cells into area between the T-cell zone and B-cell follicle in the lymph nodes [97]. The axis also promotes the migration of activated CD44^+^ CD4^+^ T cells into the central nervous system in experimental autoimmune encephalomyelitis. Taken together, 7α,25-HC is involved in both homeostasis and inflammatory cell migration [98].

#### 3.3.4. Differentiation

Oxysterols influence CD4^+^ T-cell differentiation. Naïve CD4^+^ T-cells differentiate into effector T-, or regulatory T- (Treg) cells. Effector T-cells include T helper type 1 (Th1) cells, which mainly secrete IFN-γ and defend against intracellular bacteria and tumor; Th2 cells, which secrete mainly IL-4, IL-5 and IL-13, and exert antiparasitic function; and Th17 cells that secrete mainly IL-17A and essentially defend against extracellular bacteria. Among the effector T-cells, oxysterols appear to be the most directly involved in Th17 differentiation. Desmosterol, a cholesterol precursor, and oxysterols, 7β,27-OHC and 7α,27-OHC, act as ligands of RORγt, the master transcriptional factor of Th17 differentiation, and promote the induction of human and murine Th17 cells [99,100]. Th17 cells are involved in the onset and development of autoimmune diseases. Increased levels of endogenous oxysterols binding to RORγt may promote Th17 differentiation and therefore be involved in the pathogenesis of autoimmune diseases.

To date, no evidence has shown that cholesterol and oxysterols are directly involved in Th1, Th2, and Treg differentiation though acting as ligands of their master transcriptional factors, T-bed, GATA-3, and Foxp3 [101,102,103]. However, oxysterols influence cytokine production in T-cells. Perucha et al. showed that 25-HC reduced the levels of cMaf, a key regulator of IL-10 expression, and consequently inhibited IL-10 production, that increase IFN-γ production. [103]. In a mouse model, a high cholesterol diet increased the levels of Th2 cytokine and T reg, suggesting that cholesterol may be indirectly involved in Th2 cell or Treg differentiation [104,105]. Th2 cells are the primary players in the pathogenesis of allergies. Several studies assessed the association of blood cholesterol levels with allergy onset; however, such an association has to be verified [106,107].

### 3.4. B Cells

#### 3.4.1. Activation

B cells play a central role in humoral immunity. Upon the engagement of B-cell receptor (BCR) and T-B cell interactions, B cells are fully activated and form germinal centers (GCs), which are microstructures that serve as the major sites of clonal expansion and affinity maturation of B cells, and differentiation into plasma cells in the secondary lymphoid tissue [108]. Cholesterol biosynthesis in B cells contribute to GC formation and maintenance [109]. However, the information about whether and how dietary cholesterol and oxysterol directly influence B cell activation is limited.

#### 3.4.2. Migration

Naïve B cells relocate within secondary lymphoid tissues. Upon receiving T-cell help at the border of the follicle, B-cells re-localize to interfollicular and outer follicular regions prior to their differentiation into plasmablasts or GC formation at the center of the follicle. This relocation of B cells is coordinated by an axis of 7α,25-HC and GPR183, together with an axis of CXCL13 and CCR5 [110]. Kelly et al. showed that migration of B-cells to the outer follicular regions is induced by 7α,25-HC expressing stromal cells at the perimeter of the follicle, whereas pre-GC B-cells downregulate GPR183 by BLC6 expression and are located in the center of the follicle for GC formation [111,112].

#### 3.4.3. Differentiation

Immunoglobulin (Ig)A is a major class of antibodies found in the mucosal tissues. Oxysterol 25-HC is an inhibitory factor in IgA class switching. 25-HC acts on GC B-cells in Peyer’s patches to limit differentiation of B-cells into IgA producing plasma cells by inhibiting SREBP2 activation [113]. A high-cholesterol diet enhanced 25-HC tissue concentration in Peyer’s patches and reduced the antigen-specific IgA response in mice. The results suggest that a high-cholesterol diet may reduce IgA levels and increase the risk of mucosal infection.

### 3.5. Innate Lymphoid Cells

#### 3.5.1. Migration and Inflammation

ILCs are the most recently identified leucocytes that are phenotypically similar to Th subsets but lacking TCR and BCR [114]. There are three main subsets of ILCs: ILC1, ILC2, and ILC3, which produce IFN-γ, IL-4/IL-5/IL-9/IL-13, and IL-17A/IL-22, respectively. Although ILC population is less than that of other leucocytes, the cells can produce significantly higher amounts of cytokines than T cells. ILCs play an essential role in immune homeostasis and the induction of inflammation.

Compared to other ILC subsets, ILC3 express higher level of GP183 in intestinal lamina propria. The interaction of GPR183 expressing ILC3 with 7α,25-HC secreted from fibroblastic stromal cells leads to formation of crypto patches, the clusters of lymphoid cells in the basal lamina propria of the intestines, with multiple functions, including the induction of inflammation and tissue repair [114,115]. However, increased 7α,25-HC level in the intestinal tissues causes colitis through GPR183-mediated ILC3 recruitment [114,116]. This indicates that oxysterol-GPR183 contributes to the maintenance of intestinal homeostasis and is involved in the onset of inflammation [117]. The GPR183 locus is a risk locus for inflammatory bowel diseases (IBD), suggesting a role for oxysterols in IBDs [118,119]. It is crucial to know whether and how dietary cholesterol and oxysterols influence development of ILCs and onset of IBDs.

#### 3.5.2. Differentiation

The master transcription factors for differentiation and the function of ILC1, ILC2, and ILC3, are T-bet, GATA-3, and RORγt, respectively. Oxysterols, 7β,27-OHC and 7α,27-OHC, are ligands of RORγt and may be involved in ILC3 differentiation, as observed in Th17 differentiation [99,100].

### 3.6. Impact of Cholesterol and Oxysterol on Protection from Pathogen Infection

#### 3.6.1. In Protection against Viral Infection

De novo synthesized 25-HC contributes to the defense against viral infection. Upon viral infection, macrophages and DCs, in particular plasmacytoid DCs, produce type I IFN (IFN-α and IFN-β) by recognition of virus components with TLR3, TLR7/8, and many other PRRs. Type I IFN binds to Type I IFN receptor, which is expressed in most cells in humans and rodents, and promotes the synthesis of 25-HC by inducing gene expression of cholesterol 25-hydroxylase [82]. The antiviral effects of 25-HC have been broadly indicated in experimental systems of a wide range of enveloped pathogenic viruses, e.g., VSV-SARS-CoV-2, hepatitis C virus, and Zika virus. 25-HC inhibits virus entry into cells by changing electric charge and reducing available cholesterol in the cell membranes [82,83,84,85]. In addition, 25-HC prevents virus replication by interacting with viral proteins and inducing antiviral genes and miRNA [82,83,84,85].

Type I IFN triggers several other molecules for protection against virus, in addition to 25-HC. Interferon-induced transmembrane (IFITM) 3 is one of such molecules and contributes to restrict virus entry [120]. IFITM3 interacts with membrane cholesterol via an amphipathic helix (AH) in the N terminal region of the protein, and promotes membrane fluidity, that block virus entry [121]. It is noteworthy that 25-HC and cholesterol are engaged in the prevention of virus infection by multiple mechanisms.

#### 3.6.2. In Protection against Bacterial Infection

Cholesterol-rich lipid rafts are required for a stable physical interaction of multiple pathogens with the plasma membrane in cell entry. A recent study showed that type II IFN (IFN-γ)-activated macrophages produce 25-HC and inhibit infection of *Listeria monocytogenes* and *Shigella flexneri* across epithelial cell junctions by mobilizing accessible cholesterol in the cell membrane [122]. Upon bacterial infection, macrophages and DCs produce type II IFN. As seen in the effect of type I IFN, type II IFN produces 25-HC by inducing the expression of cholesterol 25-hydroxylase [122]. 25-HC may be an effector molecule to inhibit the entry of type II-IFN inducible bacteria in the mucosal system.

There is evidence suggesting an increased risk of developing tuberculosis among people with type 2 diabetes [123]. Vrieling et al. showed that ox-LDL supports the intracellular survival of *Mycobacterium tuberculosis* in vitro by lysosomal cholesterol accumulation and subsequent dysfunction [124]. This suggests that ox-LDL create pro-bacterial infection intracellular milieu.

## 4. Cholesterol lowering Dietary Compounds

A reduction of elevated LDL cholesterol is crucial to prevent the onset and deterioration of inflammatory states and diseases associated with hyperlipidemia [125,126]. Several dietary compounds could have cholesterol lowering effect. Such dietary compounds include oat/barley β-glucans [127,128], plant sterols/stanols [129,130], omega-3 fatty acids [131,132], polyphenols [133,134], and soy proteins [135,136,137,138]. In addition to cholesterol lowering effects, dietary molecules potentially have immunomodulatory effects.

### 4.1. β-Glucans

#### 4.1.1. Cholesterol Lowering Effect

Dietary fiber changes the physical characteristics of intestinal contents that influ-ence gastric emptying, dilutes enzymes, and absorbable compounds, and modulates the digestive processes that lead to blood cholesterol and glucose attenuation [127]. The USA, EU, and several other countries, have approved oat and barley β-glucans (mostly comprising of β-1,3/1,4-glucan) to decrease blood cholesterol levels and the risk of CVD. The USFDA has approved the consumption of ≥3 g/day β-glucan from either whole oats or barley, or a combination of these, and/or ≥7 g/day of soluble fiber from psyllium seed husk to reduce CVD risk [9,10]. The European Food Safety Authority (EFSA) has ap-proved ≥3 g/day of oat β-glucan to lower blood cholesterol level [11].

The cholesterol lowering effect of oat β-glucan has been explained as the adsorp-tion of dietary cholesterol and bile acids by fiber. Furthermore, oat β-glucan increases the activity of CYP7A1, a key enzyme in the synthesis of bile acids from cholesterol, resulting in cholesterol excretion and a subsequent reduction in blood cholesterol [128,139]. Short-chain fatty acids (SCFAs) produced by the gut microbiome after β-glucan digestion could also be key molecules in decreasing cholesterol levels. Fer-mentation with human fecal bacteria in vitro and feeding animal experiments showed that oat β-glucan increases populations of Bacteroides, Prevotella and/or Roseburia species, or the family Verrucomicrobia that produce SCFA such as acetate, propionate, and bu-tyrate, in the gut microbiome [128]. These SCFAs promote bile acid synthesis from cholesterol in the intestines and liver, thus reducing total cholesterol levels.

#### 4.1.2. Immunomodulatory Effects

In addition to the cholesterol lowering effect, the immunomodulatory effect of oat and barley β-glucan has been of concern. Beta-1,3- and β-1,3/1,6-glucans derived from fungi or bacteria stimulate macrophages and DCs [15,140]. When such glycans bind to TLR2 and Dectin-1, they trigger pro-inflammatory cytokine production, cell maturation, and migration of APCs, and elicit Th1 or Th17 cell-biased adaptive immune responses [15,140]. In contrast, β-1,3/1,4-glucans derived from cereals do not bind PRRs, as β-1,3- and β-1,3/1,6-glucans. However, β-1,3/1,4-glucans may have immunological function. Pan et al. showed that oat glucan induces the metabolic reprogramming of murine monocytes and macrophages by activating the glycolytic pathway and contribute to establish trained immunity, i.e., innate immune memory responding higher to the second microbiome components [141].

Oat/barley β-glucan digested by the gut microbiome might induce immune mod-ulation. A high-fiber diet reshapes the gut microbial ecology, increases SCFA release, and reduces inflammation in murine models of inflammatory disorders [142,143]. SCFA enhances histone H3 acetylation in the promoter region of Foxp3 and promotes Treg differentiation [144]. SCFA also promotes IgA class switch in B cells and their differen-tiation into plasma cells [145]. IgA antibodies regulate the diversity of microbiome and contribute to keep bacterial community, which in turn maintains the expansion of Foxp3+ CD4+ T-cells furthermore and balances to mediate host-microbe symbiosis [143,146]. Collectively, oat/barley β-glucans digested by the gut microbiome might re-duce cholesterol-associated inflammation and enhance mucosal defense.

### 4.2. Plant Sterols and Stanols

#### 4.2.1. Cholesterol Lowering Effects

Plant sterols and stanols are natural products that can be ingested through vegetables and other plants. Plant sterols are structurally similar to cholesterol and act as cholesterol-absorption inhibitors by displacing cholesterol from bile emulsions in the intestine [129,130]. They also suppress cholesterol biosynthesis by inhibiting the expression of HMG-CoA reductase and SREBP-2 [147]. Stanols are hydrogenation compounds of plant sterols. They occur in nature at a lower abundance than plant sterols and decrease cholesterol levels via similar mechanisms.

The dietary intake of plant sterols and stanols is usually between 150–350 and 50 mg/day in adults, which is lower than the amount required to reduce blood cholesterol levels [148]. The USFDA has approved the consumption of ≥ 1.3 g/day of plant (or vegetable oil) sterol esters or ≥ 3.4 g/day of plant (or vegetable oil) stanol esters to reduce risk of heart disease [13]. The EFSA approved a daily intake of 2.0–2.4 g of phytosterols added to appropriate fat-based and low-fat foods such as milk and yoghurt to reduce blood cholesterol [14].

#### 4.2.2. Immunomodulatory Effects

The primary components of plant stanols are sitostanol and campestanol. Anti-inflammatory effect of sitostanol was shown in in vitro cell culture systems, e.g., inhibition on LPS-induced pro-inflammatory events by reductions in gene and protein expression of TLR4, MyD88 and IL-1 receptor associated kinase (IRAK)-1 (Figure 4) [149,150]. It suggests that sitostanol potentially inhibit ox-LDL-mediated inflammatory responses in monocytes and macrophages, since TLR4 is a receptor of ox-LDL. In clinical studies, the consumption of plant stanol ester shifted toward from Th2 to Th1 type immune response in PBMCs from patients with asthma [151], that might be beneficial to reduce the disease state of allergy. The reduction effects of plant stanols on the serum levels of TNF-α were observed in healthy subjects and patients with inflammatory disorders, but only inconsistently [152,153]. Pentalinonsterol (cholest-4,20,24-trien-3-one), a sterol from *Pentalinon andrieuxii*, enhanced phospholipase A2 (PLA2) activity, and TNF-α production in BMDMs [154]. Thus, not all plant sterols appear to be anti-inflammatory compounds. The impact of oxidized plant sterols/stanols should also be investigated for a better understating of health effects of these dietary molecules.

### 4.3. Omega-3 Fatty Acids

#### 4.3.1. Cholesterol Lowering Effects

Omega-3 fatty acids include eicosapentaenoic acid (EPA), docosahexaenoic acid (DHA), and alpha-linolenic acid (ALA). Both EPA and DHA are abundant in fish oils, whereas ALA is mainly found in plant derived oils. ALA must be converted to EPA to exert equivalent biological effects, but the conversion of ALA to EPA and DHA is limited. The USA, several countries in Europe, and other regions, have approved prescriptions of omega-3 fatty acid products, icosapent ethyl (EPA ethyl ester) and omega-3-acid ethyl esters (ethyl ester of EPA and DHA), as adjuncts to a nutritious diet to treat hypertriglyceridemia by reducing triglyceride and LDL levels [155]. The proposed action mechanisms of omega-3 fatty acid products include the inhibition of diacylglycerol acyltransferase, increased plasma lipoprotein lipase activity, decreased hepatic lipogenesis, and increased hepatic β-oxidation [131,132].

In 2019 the USFDA only qualified the claim that food and dietary supplements containing ≥ 0.8 g/day of EPA and DHA might reduce the risk of hypertension and coronary heart disease, due to inconsistent and inconclusive evidence of clinical studies [156,157]. The EFSA did not approve the heath claims of EPA and DHA related with neither maintenance of normal blood LDL nor HDL-cholesterol concentrations nor cholesterol lowering effect [158].

#### 4.3.2. Immunomodulatory Effects

EPA and DHA exert anti-inflammatory effects in in vitro culture systems of APCs and rodent models of inflammatory diseases [20,159]. Macrophages and DCs express GPR 120 binding EPA and DHA [160]. GRP120 engagement inhibits LPS-stimulated NF-κB-JNK cascade and activation of NLRP3 inflammasomes, thus suppressing pro-inflammatory cytokine production [160] (Figure 4). Both EPA and DHA also regulate activities of multiple transcriptional factors including NF-κB, SREBP, and PPARs, and kinase mechanistic target of rapamycin (mTOR) and related gene expression [21,161]. Considering these actions of DHA and EPA in intracellular events, it could be postulated that these omega-3 lipids inhibit oxysterol-induced inflammatory responses in APCs.

Importantly, the metabolites of EPA and DHA, resolvins, protectins, and maresins, also exsert anti-inflammatory effects [162,163]. The effects of such metabolites on innate immunity include increasing phagocytosis and limiting activation and trafficking of macrophages and neutrophils [164,165,166,167]. Resolvins, protectins, and maresins, also influence on adaptive immunity e.g., by promoting de novo generation and function of Foxp3^+^ Treg and the differentiation of human B cells to antibody-secreting cells via GPR32 [168,169].

The anti-inflammatory effects of EPA and DHA on rheumatoid arthritis, IBD, and other inflammatory disorders, have been clinically assessed. The potential effects of omega-3 fatty acids on mobility reduction and severe acute respiratory syndrome coronavirus 2 (SARS-CoV-2) have also been of great concern [170]. However, the results of these clinical studies have been inconsistent and inconclusive, perhaps due to differences in populations with various grades of inflammation, the analyses of different types of cells, and the quality and quantity of EPA and DHA. Endogenous levels of EPA and DHA and gut microbiome profiles could also influence the effects of exogenous EPA and DHA. Further studies are required to identify the physiological conditions under that omega-3 fatty acids exert anti-inflammatory effects.

### 4.4. Polyphenols

#### 4.4.1. Cholesterol Lowering Effects

Polyphenols are secondary metabolites containing one or more hydroxyl groups. Studies have shown that many polyphenols such as catechins, gallic acid, quercetin, chlorogenic acid, curcumin, and resveratrol, lower cholesterol levels in mice and in cultured cells in vitro [133,134].

Polyphenols have diverse mechanisms of action. For instance, (i) catechins reduce cholesterol absorption in intestines by displacing cholesterol from intestinal micelles [171,172]. (ii) Curcumin and luteolin reduce cholesterol absorption by expressing Niemann-Pick C1-Like 1 (NPC1L1), a critical protein in the absorption, and several other molecules in intestinal epithelial cells [173,174]. (iii) Berberine, curcumin, epigallocatechin gallate, resveratrol, and many others, increase the expression of hepatic LDL receptors by inhibiting the expression of proprotein convertase subtilisin/kexin type 9 (PCSK9), which is involved in cholesterol homeostasis [175,176]. (iv) Genistein inhibits the processing and the nuclear translocation of SREBP2, a regulator of cellular cholesterol synthesis, whereas epigallocatechin gallate, luteolin, and several other polyphenols, inhibit expression of SREBP2 (Figure 2) [133,134]. (v) Quercetin, epigallocatechin gallate, naringin, and several others, increase the activity of CYP7A1, the rate-limiting enzyme that promotes cholesterol-to-bile acid conversion in the classical pathway of bile acids [177]. (vi) Anthocyanins reduce biliary cholesterol levels by increasing the expression of the ATP-binding cassette (ABC) transporters G5 (ABCG5), G8 (ABCG8), and decreasing the expression of ACAT2, in the liver and small intestine [133,134].

Importantly, molecular mechanisms for the cholesterol lowering effect of polyphenols have been elucidated mainly in experimental systems. After ingestion, the presence of dietary polyphenols in the circulatory system rarely exceeds nanomolar concentrations in humans [133,178]. This inefficient absorption and low levels of circulation of polyphenols could explain the inconsistent lowering effects of polyphenols on blood cholesterols in clinical studies [179,180,181]. Neither the USFDA nor the EFSA has approved polyphenols for reducing blood cholesterol levels.

#### 4.4.2. Immunomodulatory Effects

Several polyphenols exert anti-inflammatory effects by inhibiting the expression of pro-inflammatory cytokines, cyclooxygenase (Cox)-2, and prostaglandin E2 (PGE2) in macrophages and DCs, and inducing Treg differentiation in experimental systems [17,182]. The anti-inflammatory effects of polyphenols on APCs are implemented by diverse mechanisms (Figure 4). Inflammatory stimuli, e.g., TLR4 engagement, trigger the overproduction of ROS, that promote activation of NF-кB, MAPK, and PI3K/Akt-mediated pathways and gene expression of pro-inflammatory cytokines, whereas polyphenols reduce ROS generation [183]. Multiple polyphenols inhibit phosphorylation and the degradation of the inhibitor of NF-κB (IκB) and/or phosphorylation of IKK (IκB kinase), due to their antioxidant properties [182].

In addition to the effects on IκB and IKK, catechins induce hypoacetylation in the p65 subunit of NF-κB (RelA) by inhibiting the activity of histone acetyltransferase (HAT) enzymes, that inhibits the transcriptional activity of NF-κB [184,185]. Resveratrol upregulates activity of SIRT1, a nicotinamide adenine dinucleotide (NAD)-dependent deacetylase, that also induces the hypoacetylation of RelA [186]. Polyphenols inhibit formation of NLRP3-inflammasome and thereby reduce IL-1β production in APCs [183,184,185]. Furthermore, polyphenols act as the natural ligands of nuclear factor erythroid 2-related factor 2 (NRF2), that negatively regulates the gene expression of pro-inflammatory cytokines [183,184,185]. 

Polyphenols influence T-cell activation and differentiation. Epigallocatechin gallate inhibits Zeta-chain-associated protein kinase 70 (ZAP-70), thus suppressing CD3-mediated T-cell receptor signaling [187]. Epigallocatechin gallate also acts as a DNA methyltransferase (DNMT) inhibitor. The inhibition of DNMT by epigallocatechin gallate induces Foxp3 expression, which leads to an increased number of Tregs in mice [188]. Curcumin, resveratrol, and several other polyphenols inhibit Th17 differentiation and promote Treg differentiation by reducing RORγt mRNA expression and enhancing Foxp3 expression [189]. Polyphenols activate AMP-activated protein kinase (AMPK), and suppress signal transducer and the activator of transcription 3 (STAT3), hypoxia inducible factor 1 alpha (HIF-1α), and the mechanistic target of rapamycin (mTOR)-mediated pathways, that could also influence T-cell activation and differentiation [189]. Naringin, a flavonoid, induces Treg differentiation through pathways involving the aryl hydrocarbon receptor [190].

The influence of polyphenols on the development of ILCs has not been investigated well. The master transcription factors of ILC1, ILC2, and ILC3, are T-bet, GATA-3, and RORγt, respectively [191]. Several polyphenols affect RORγt expression [189]; hence, those might inhibit the oxysterol-induced differentiation of Th17 cells and ILC3.

Dietary polyphenols have been extensively investigated for their immunomodulatory effects in murine models of immune disorders, e.g., rheumatoid arthritis, IBD, and allergies [191,192,193,194], but the effects in humans have been less investigated. A meta-analysis of randomized controlled trials showed that resveratrol could decrease serum levels of TNF-α and IL-6 but not IL-1 and IL-8 [195]. A meta-analysis of randomized controlled trials assessing the effect of curcumin described a significant decrease of IL-1 and TNF-α, and non-significant decrease of IL-6 and increase of IL-8 levels. A recent meta-analysis of polyphenols on the treatment of IBD suggested the potential effect of curcumin and resveratrol on gastrointestinal symptom and remission [196,197]. However, it is still not clear how polyphenols exsert anti-inflammatory mechanisms, whether inhibiting pro-inflammatory events in immune cells, or through different mechanisms of action in humans. Gut microbiota could influence the effects of polyphenols due to its ability for producing bioactive metabolite from polyphenols, suggesting that an individual’s microbiota profile is a factor to exsert immune modulation of polyphenols.

### 4.5. Soy Proteins

#### 4.5.1. Cholesterol Lowering Effects

The USFDA approved the consumption of ≥ 25 g/day of soy proteins to reduce the risk of coronary heart disease in 1999, based on their observed ability to decrease serum cholesterol levels [198]. However, the USFDA proposed the revoking of the approval in 2017, with reference to inconsistent findings of the effects of soy protein since 1999 [199]. The USFDA only qualified the health claim for soy proteins in 2019, due to supportive, but inconclusive evidence, derived from a meta-analysis of 46 randomized controlled trials [200]. The EFSA declined an application for a health claim of cholesterol lowering effect of isolated soy protein (ISP), due to a lack of a cause-and-effect relationship between ISP and a reduction in blood LDL-cholesterol concentrations [201].

The main constituents of soybean proteins are β-conglycinin and glycinin, which comprise 65%–80% of the protein fraction, or 25%–35% of the seed weight. Several studies on rodent models showed that purified soy proteins or soy protein-enriched products lower cholesterol [202,203]. Beta-conglycinin reduced blood cholesterol by increasing the levels of fibroblast growth factor 21 (FGF21) in mice [202]. A diet containing soy proteins stimulated bile acid excretion in the liver and intestines of rats by repressing FGF15 and small heterodimer partner (SHP), that reduced cholesterol levels [203]. Soy protein-derived peptides might alter expression of the transcription factors SREBP2, PPARs, and HIF1α, and enzymes such as CYP7A1 and CYP8B1, that are involved in bile acid synthesis from cholesterol [137,138]. Studies suggest that plant proteins may affect cholesterol and bile acid metabolism through gut microbiome [204]. However, the ability of soy proteins to exert cholesterol-lowering mechanisms in humans awaits verification.

#### 4.5.2. Immunomodulatory Effects

The major allergens in soy are β-conglycinin (7S globulin) and glycinin (11S globulin) [205]. To date, eight major soy proteins, including β-conglycinin (allergen Gly m 5) and glycinin (allergen Gly m 6), have been registered in the systematic allergen nomenclature approved by the World Health Organization and International Union of Immunological Societies (WHO/IUIS) Allergen Nomenclature Sub-committee.

Beta-conglycinin is the closest known homolog of peanut 7S globulin, which is the major peanut allergen, Ara h 1 [206]. Natural Ara h 1 carries high mannose (Man5GlcNAc2 and Man6GlcNAc2) and nonfucosylated complex N-glycans (Man4XylGlcNAc2 and Man3XylGlcNAc2) [207]. Studies have shown that such carbohydrate residues bind to DC-specific intercellular adhesion molecule-3 grabbing nonintegrin (DC-SIGN) or Scavenger receptor A (SR-A) and thus activate human monocyte-derived DC and macrophages [208,209]. Beta-conglycinin also carries N-linked glycans comprising 77% mannose and 23% N-acetylglucosamine, and thus might have the intrinsic immunogenicity to stimulate macrophages and DCs like peanut 7S globulin [210].

Although β-conglycinin and glycinin have potential allergenicity, soy protein-derived peptides may exert immunomodulatory function. Yi et al. showed that peptide derived from protease-treated crude soy proteins reduces LPS-induced pro-inflammatory cytokine expression by the inhibition of NF-kB pathway in a murine macrophage cell line [211]. Interestingly, Rein et al. showed the application of artificial intelligence to identify anti-inflammatory peptides from rice proteins, which inhibit IL-6-mediated inflammatory responses [212]. The application of artificial intelligence to identify functional peptides would promote the utilization of soy protein-derived peptides for immune modulation as well as other health benefits including cholesterol lowering effects.

## 5. Conclusions

High levels of extracellular cholesterol and oxysterols skew the immune system toward a pro-inflammatory state, whereas homeostatic levels of cholesterol biogenesis and de novo synthesis of oxysterols are essential for the differentiation, migration, and function of immune cells. A high cholesterol diet could negatively affect the development of inflammatory disease, whereas cholesterol lowering dietary molecules could help to reduce excess biogenesis of cholesterol and suppress inflammatory milieu subsequently.

Notably, the oxidative products of cholesterol are not only oxysterols. The reaction of cholesterol 5α,6α-epoxide with peptide-bound lysine has been shown in a model system at conditions reflecting cooking or frying processes [213]. The lipation of proteins and peptides by cholesterol (ep)oxides during food processing and should, therefore, be possible. Beneath oxidized sterols, oxidized phytosterols also play a role in plant-based foods [214]. In different cold-pressed plant oils, 8–35 mg/kg total phytosterol oxidation products were determined and 5–110 mg/kg in refined oils [30]. The impact of such oxidized products on the immune system has to be further warranted. It is further essential to elucidate the impact of lipid-modulating dietary components on the immune system and in the treatment of specific inflammatory and infectious diseases.

Dietary compounds with cholesterol lowering effects could affect events of immune cells by interfering in signaling pathways and gene expressions and exert immune modulation [17,20,215]. Dietary compounds also alter gut microbiome profile and induce metabolite production from the microbiome, that could contribute to immune modulation. Comprehensive studies about potential regulatory pathways are necessary to understand and gain overall beneficial effects of such dietary compounds.

## Figures and Tables

**Figure 1 ijms-23-12236-f001:**
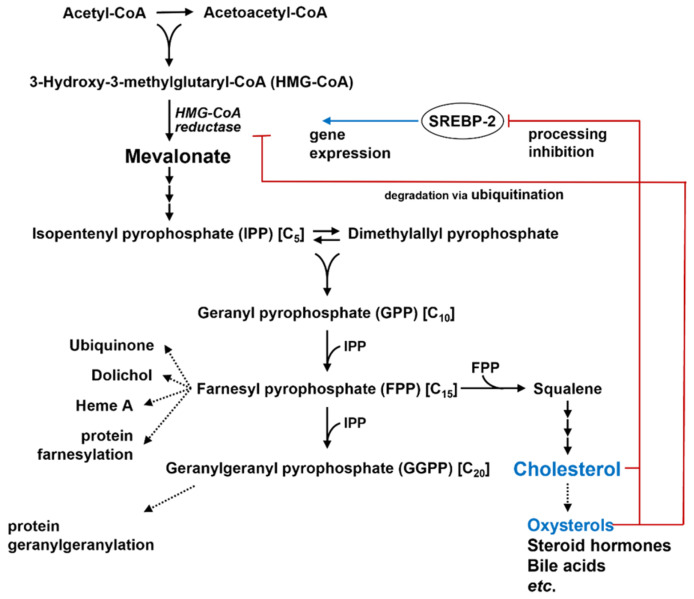
The major pathway for de novo cholesterol synthesis. Cholesterol is synthesized through the mevalonate-isoprenoid pathway. A series of isoprenoid products of this pathway play vital roles in multiple cellular functions. SREBP: sterol regulatory element-binding protein.

**Figure 2 ijms-23-12236-f002:**
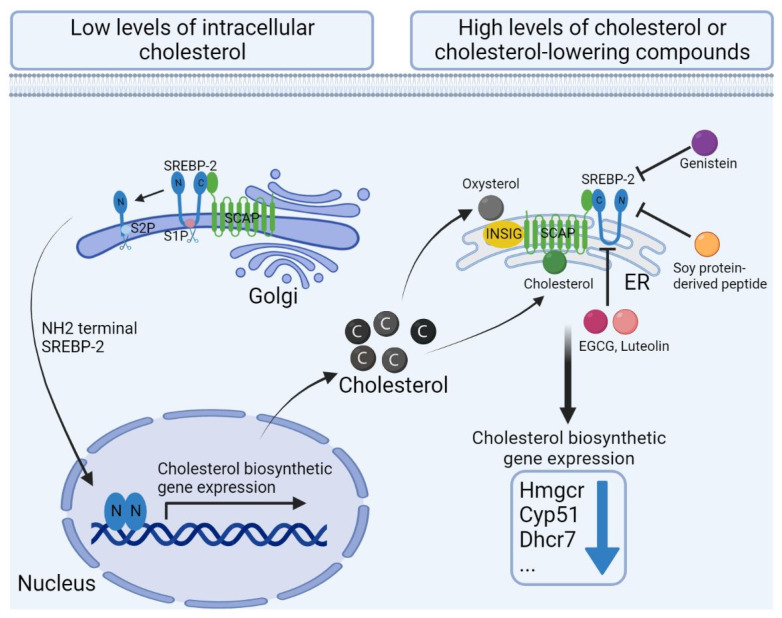
The role of sterol regulatory element-binding protein (SREBP)/SREBP-cleavage-activating protein (SCAP) in the regulation of cholesterol synthesis. When the levels of intracellular cholesterol are low, SREBP-2/SCAP complex is transported from endoplasmic reticulum (ER) to the Golgi apparatus and processed to release the NH2-terminal fragment of SREBP-2 (nSREBP-2). nSREBP-2 is translocated into the nucleus to induce expression of genes related with mevalonate pathway, that increasing cholesterol synthesis. When the levels of intracellular cholesterol are high, SCAP forms a complex with insulin-induced gene 1 (INSIG1) and SREBP-2, and remains inactive in the ER. Oxysterols bind INSIG, that also leads to form the SREBP-2/SCAP/INSIG complex and thereby prevents SREBP-2 processing. Several polyphenols, e.g., epigallocatechin gallate (EGCG) and luteolin, and soy protein derived peptides, could inhibit SREBP2 expression, reducing cholesterol synthesis. A soy-derived flavonoid, genistein, inhibits the processing and the nuclear translocation of SREBP2.

**Figure 3 ijms-23-12236-f003:**
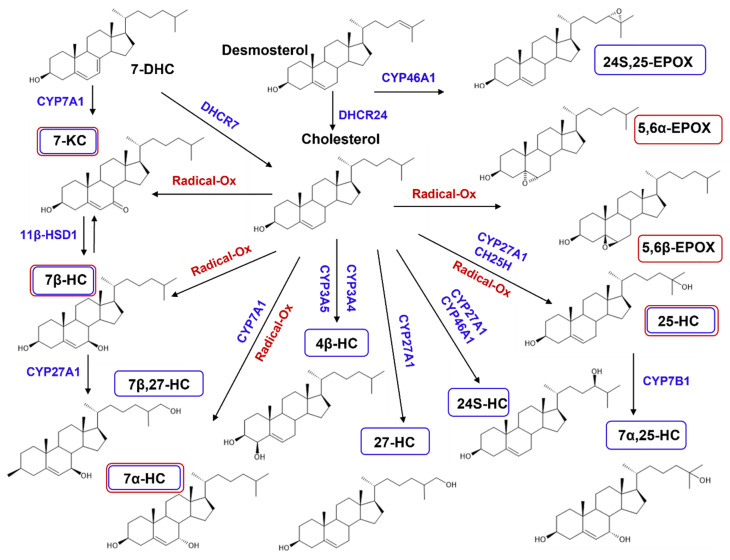
Pathways of cholesterol oxidation. Cholesterol gives rise to oxysterols via enzymatic and non-enzymatic routes. Dietary oxysterols are produced by the oxidation of cholesterol with oxygen and ROS, i.e., non-enzymatic routes. De novo synthesized oxysterols are produced by enzymes belonging to the cytochrome P450 (CYP) family. CH25: cholesterol 25-hydroxylase, EPOX: epoxycholesterol, DHCR: dehydrocholesterol reductase, HC: hydroxycholesterol, HSD: hydroxysteroid dehydrogenase, KC: ketocholesterol, Radical-Ox: radical oxidation.

**Figure 4 ijms-23-12236-f004:**
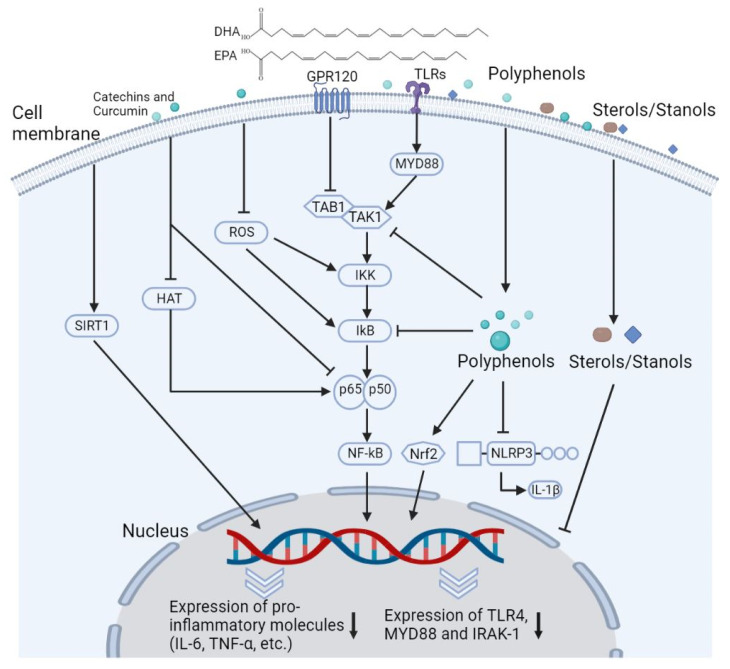
Possible effects of plant sterols/stanols, omega-3 fatty acids, and polyphenols, on cellular events in antigen presenting cells. Toll-like receptor (TLR) engagement triggers pro-inflammatory responses in macrophages and dendritic cells. Plant sterols/stanols inhibit gene expression of TLR4, MyD88, and IL-1 receptor associated kinase (IRAK)-1. Eicosatetraenoic acid (EPA) and docosahexaenoic acid (DHA) bind G protein-coupled receptor (GPCR) 120. GPR120 engagement inhibits TLR4-induced association of TGF-β-activated kinase 1 (TAK1)-binding protein 1 (TAB1) with TAK1, that block nuclear factor-kappa B (NF-kB)-mediated pathway. Polyphenols inhibit NF-kB-mediated pathways by inhibiting phosphorylation of IκB kinase (IKK) and the degradation of inhibitor of NF-κB (IκB) and altering activity of enzymes including Sirtuin 1 (SIRT1) and histone acetyltransferase (HAT). Polyphenols also inhibit formation of NLR Family Pyrin Domain Containing 3 (NLRP3) inflammasome. Polyphenols are ligands of nuclear factor erythroid 2-related factor 2 (Nrf2), that negatively regulates gene expression of pro-inflammatory cytokines.

## Data Availability

Not applicable.

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
