# Peer review of "The Impacts of Cholesterol, Oxysterols, and Cholesterol Lowering Dietary Compounds on the Immune System"

_ijms, 2022, doi:10.3390/ijms232012236_

Round 1

Reviewer 1 Report

This manuscript “The impacts of cholesterol, oxysterols, and cholesterol lowering dietary compounds on the immune system” fits well with the Aims and Scope of the International Journal of Molecular Science.  In this manuscript Dr. Yanagisawa et al discussed the knowledge intake and de novo synthesis of cholesterol and oxysterols and the detail impacts of the cholesterol and oxysterols on the various immune cells, as well the possible immunomodulatory mechanism elicited by cholesterol-lowing dietary compounds. This manuscript has been written well, however some important information is missing.

The major comments are as below:

1.    In the part of 2.2.2 De novo synthesized oxysterols: Numerous articles have shown that 25HC is more important than 27-HC in many biological processes. The biosynthesis, distribution, and function of 25-hydroxycholesterol should be discussed in more detail. Both 25-HC and 27-HC are synthesized in mitochondria, which should be mentioned.

2.     In the part of 3. The impact of cholesterol and oxysterols on immune cells: There is no evidence to show that cholesterol affect the immune response in immune cells, which does not fit the title.

3.    In the part of 3.1.2 Induction of anti-inflammation: Recent reports have shown that the function of global regulation on the inflammation by oxysterols, most likely, via epigenomic modification and epigenetic regulation, which maybe more important than sever as LXRs ligands. This should be discussed or mentioned.

4.    In Reference: the format of reference 11, 12, 14, 143, and 185 should be consistent.

Author Response

Comment 1
1.    In the part of 2.2.2 De novo synthesized oxysterols: Numerous articles have shown that 25HC is more important than 27-HC in many biological processes. The biosynthesis, distribution, and function of 25-hydroxycholesterol should be discussed in more detail. Both 25-HC and 27-HC are synthesized in mitochondria, which should be mentioned.

Reply 1

Thank you very much for valuable comments. We added texts about the biosynthesis, distribution, and function of 25-hydroxycholesterol in the revised manuscript (please see lines 176-186 in pages 4-5, and lines 458-468 in page 11 in the manuscript with marked correction). We also described that 25-HC and 27-HC are synthesized in mitochondria (lines 187-194 in page 5). The texts about that the role of mitochondrial sterol 27-hydroxylase in production of bile acid is also described (lines 190-199 in page 5).  

Comment 2
 2.     In the part of 3. The impact of cholesterol and oxysterols on immune cells: There is no evidence to show that cholesterol affect the immune response in immune cells, which does not fit the title.

Reply 2

In section 3.3. (3.3.1. Activation), we described that (i) cholesterol is a key component in T-cell activation and effector function, and (ii) cholesterol induces T cell exhaustion in tumor environment (lines 345-363, page 9). Therefore, we would like to keep the title.

Comment 3
 3.    In the part of 3.1.2 Induction of anti-inflammation: Recent reports have shown that the function of global regulation on the inflammation by oxysterols, most likely, via epigenomic modification and epigenetic regulation, which maybe more important than sever as LXRs ligands. This should be discussed or mentioned.

Reply 3

Following the reviewer’s suggestion, we added texts about the function of global regulation on the inflammation by oxysterols via epigenomic modification and epigenetic regulation. Please see lines 273-289 in page 7.

Comment 4
 4.    In Reference: the format of reference 11, 12, 14, 143, and 185 should be consistent.

Reply 4

We corrected the reference format. Please see revised reference 11, 12, 14, 158, and 201 in the revised manuscript.

Reviewer 2 Report

Yanagisawa et al. provide a very comprehensive review of how cholesterol and oxysterols contribute to immunity on multiple levels. This exhaustive review will provide an important reference for many different fields and is much appreciated for that reason. It is well written and easy to follow for both specialists and non-specialists. However, I have a couple of requests for modification, which will improve the manuscript greatly.

Major:

1.     It would be helpful if another figure was added that highlighted the functions of SREBP proteins, since they are referred to in many different sections of the manuscript. While the authors indicate clearly its role as a transcription factor and one that induces the transcription of genes involved in cholesterol synthesis, can the authors discuss some of those downstream genes and how they fit into the overall cholesterol homeostasis pathways? In this new figure, the authors could also place known inhibitors of SREBP, which are also mentioned in the text at various places.

2.     The authors divide the manuscript into many different sections, including the roles played by cholesterol/oxysterols in different immune cell types. However, cholesterol/oxysterol influence immunity in other ways, in essentially all cell types of the body. For example, cholesterol and oxysterols are important for providing intrinsic defense against virus entry. This concept is alluded to on page 6, which is much appreciated. However, since most cells in the body can respond to exogenous interferons, most cells can induce CH25H and produce 25-HC to impose a barrier to virus entry. Therefore, it is misleading that these discussions are limited to dendritic cells. It would be better to create a new section of the manuscript that discusses the roles of cholesterol/oxysterols on intrinsic immunity, and to make this section independent of certain cell types. Majdoul et al. NRI, 2022 review the intrinsic defense genes CH25H and IFITM3 in the context of antiviral defense and discuss the antiviral roles of 25-HC in depth, and so this should be cited and discussed. More recently, IFITM3 has been shown to bind to cholesterol in order to fulfill its antiviral properties (inhibition of virus entry) (Rahman et al. JMB, 2022) and this could be discussed as other ways that cholesterol/oxysterols directly contribute to immune defense.

Minor:

1.     Abstract: Remove comma before “that plays a significant…”

2.     Page 5: PRRs is misspelled as “PPRs”

Author Response

Comment 1
It would be helpful if another figure was added that highlighted the functions of SREBP proteins, since they are referred to in many different sections of the manuscript. While the authors indicate clearly its role as a transcription factor and one that induces the transcription of genes involved in cholesterol synthesis, can the authors discuss some of those downstream genes and how they fit into the overall cholesterol homeostasis pathways? In this new figure, the authors could also place known inhibitors of SREBP, which are also mentioned in the text at various places.

Reply 1

Thank you very much for your valuable comments. We added a figure and texts explaining for functions of SREBP proteins, SREBP-induced genes involved in cholesterol homeostasis pathways, and inhibitors of SREBP (e.g. oxysterol, polyphenol, soy protein-derived peptides). Please see figure 2 in page 3, and lines 78-90 in page 2 of the revised manuscript with marked correction.

Comment 2
The authors divide the manuscript into many different sections, including the roles played by cholesterol/oxysterols in different immune cell types. However, cholesterol/oxysterol influence immunity in other ways, in essentially all cell types of the body. For example, cholesterol and oxysterols are important for providing intrinsic defense against virus entry. This concept is alluded to on page 6, which is much appreciated. However, since most cells in the body can respond to exogenous interferons, most cells can induce CH25H and produce 25-HC to impose a barrier to virus entry. Therefore, it is misleading that these discussions are limited to dendritic cells. It would be better to create a new section of the manuscript that discusses the roles of cholesterol/oxysterols on intrinsic immunity, and to make this section independent of certain cell types. Majdoul et al. NRI, 2022 review the intrinsic defense genes CH25H and IFITM3 in the context of antiviral defense and discuss the antiviral roles of 25-HC in depth, and so this should be cited and discussed. More recently, IFITM3 has been shown to bind to cholesterol in order to fulfill its antiviral properties (inhibition of virus entry) (Rahman et al. JMB, 2022) and this could be discussed as other ways that cholesterol/oxysterols directly contribute to immune defense.

Reply 2

We created a new section that discusses the roles of cholesterol/oxysterols in intrinsic immunity (lines 458-488 in pages 11-12). We also referred two reviews (Majdoul et al. NRI, 2022 and Rahman et al. JMB, 2022; references 120 and 121 in page 20) and discussed the anti-viral properties of 25-HC and IFITM3 (lines 469-474 in page 11).

Comment 3

Abstract: Remove comma before “that plays a significant…”

Reply 3

We removed the comma.

Comment 4

Page 5: PRRs is misspelled as “PPRs”

Reply 4

We corrected the spell (line 249, page 7).

Reviewer 3 Report

The article entitle “The impacts of cholesterol, oxysterols, and cholesterol lowering dietary compounds on the immune system” By Yanagisawa e col. described the effects in macrophages, dendritic cells (DCs), and innate lymphoid cells (ILCs) in innate immunity, and T cells and B cells in adaptive immunity by cholesterol, oxysterol and cholesterol lowering compounds. The article after the description of oxysterol and cholesterol synthesis describe how they can affect different cells of immune system as macrophages, dendritic cells, T cells, B cells and Innate lymphoid cells. Some cholesterol lowering dietary compounds are described and correlated to possible immunomodulatory effects.  This is an important topic and the paper brings together many pieces of information regarding to a high biological activity of oxysterols.

 Minors:

1- The item “3.4.1. Activation

T-cell activation is initiated by the engagement of the T-cell antigen receptor (TCR)/CD3 complex. Cholesterol is a key component of membrane lipids, and is critical for TCR clustering and signaling, and T-cell function [80]. SREBP2-mediated pathway inducing cholesterol synthesis was shown to be dispensable for proliferation and effector function of CD8+ T cells [81], whereas LXR-mediated pathway inducing cholesterol efflux negatively regulated T cell activation [82].”

The paragraph is very confuse and difficult of understanding

2- Described in the text better the figure 3 and improve the description in the legend to make it more clear

 Suggestion:

1 - Inside the item dietary and de novo synthesized oxysterols described also the ox-LDL. It is cited in the activation of Macrophages, dendritic cells, Plant sterols and stanols and Omega 3.

2- It may be helpful to summarize all the immunomodulatory effects of lowering dietary compounds described in text in a table

Author Response

Comment 1

The article entitled “The impacts of cholesterol, oxysterols, and cholesterol lowering dietary compounds on the immune system” By Yanagisawa e col. described the effects in macrophages, dendritic cells (DCs), and innate lymphoid cells (ILCs) in innate immunity, and T cells and B cells in adaptive immunity by cholesterol, oxysterol and cholesterol lowering compounds. The article after the description of oxysterol and cholesterol synthesis describe how they can affect different cells of immune system as macrophages, dendritic cells, T cells, B cells and Innate lymphoid cells. Some cholesterol lowering dietary compounds are described and correlated to possible immunomodulatory effects. This is an important topic and the paper brings together many pieces of information regarding to a high biological activity of oxysterols.

Reply 1

Thank you very much for evaluating that our review article deals with an important topic.

Comments 2

Minors: The item “3.4.1. Activation

T-cell activation is initiated by the engagement of the T-cell antigen receptor (TCR)/CD3 complex. Cholesterol is a key component of membrane lipids, and is critical for TCR clustering and signaling, and T-cell function [80]. SREBP2-mediated pathway inducing cholesterol synthesis was shown to be dispensable for proliferation and effector function of CD8+ T cells [81], whereas LXR-mediated pathway inducing cholesterol efflux negatively regulated T cell activation [82].” The paragraph is very confused and difficult of understanding.

Reply 2

We rephrased these texts to be understandable. Please see it in lines 345-363, page 9 in the revised manuscript with marked correction.

Comments 3

Described in the text better the figure 3 and improve the description in the legend to make it more clear

Reply 3

We rephrased the texts for figure 3 and improved description in the legend to make it clear (lines 662-682 in page 16, and lines 716-725 in page 17).

Comment 4

Suggestion 1 - Inside the item dietary and de novo synthesized oxysterols described also the ox-LDL. It is cited in the activation of Macrophages, dendritic cells, Plant sterols and stanols and Omega 3.

Reply

Oxysterols are contained in ox-LDL abundantly and contribute to induction of inflammatory immune responses. This is the reason why we described about the inflammatory properties of oxidized low-density lipoprotein (ox-LDL) in several sections. We had added texts that oxysterols contained in ox-LDL in lines 252-253, page 7 in the revised manuscript.

Suggestion 2- It may be helpful to summarize all the immunomodulatory effects of lowering dietary compounds described in text in a table.

Reply

We summarized the immunomodulatory effects of polyphenols, omega-3 fatty acids and plant sterols and stanols in legend of Fig. 3 (lines 716-724, page 17), instead of making a Table. The possible immunomodulatory effects of several polyphenols and soy protein derived peptides on SRBRP-2 is indicated in legend of Fig. 2 (lines 122-130, page 3). Oat glucan does not possess immunomodulatory effects. Soy proteins contain rather immunostimulatory molecules. Thus, we did not include it in the legends.